# Effects of *Panax notoginseng* Saponins Encapsulated by Polymerized Whey Protein on the Rheological, Textural and Bitterness Characteristics of Yogurt

**DOI:** 10.3390/foods13030486

**Published:** 2024-02-02

**Authors:** Zengjia Zhou, Huiyu Xiang, Jianjun Cheng, Qingfeng Ban, Xiaomeng Sun, Mingruo Guo

**Affiliations:** 1Key Laboratory of Dairy Science, College of Food Science, Northeast Agricultural University, Harbin 150030, China; m15255825668@163.com (Z.Z.); 13534253985@163.com (H.X.); jjcheng@neau.edu.cn (J.C.); qfban@neau.edu.cn (Q.B.); 2Department of Nutrition and Food Science, College of Agriculture and Life Sciences, University of Vermont, 351 Marsh Life Science Building, 109 Carrigan Drive, Burlington, VT 05405, USA

**Keywords:** *Panax notoginseng* saponin-polymerized whey protein nanoparticles, yogurt, rheological characteristics, textural characteristics, bitterness

## Abstract

*Panax notoginseng* saponins (PNSs) have been used as a nutritional supplement for many years, but their bitter taste limits their application in food formulations. The effects of PNS (groups B, C, and D contained 0.8, 1.0 and 1.2 mg/mL of free PNS, respectively) or *Panax notoginseng* saponin-polymerized whey protein (PNS-PWP) nanoparticles (groups E, F, and G contained 26.68, 33.35 and 40.03 mg/mL of PNS-PWP nanoparticles, respectively) on the rheological, textural properties and bitterness of yogurt were investigated. Group G yogurt showed a shorter gelation time (23.53 min), the highest elastic modulus (7135 Pa), higher hardness (506 g), higher apparent viscosity, and the lowest syneresis (6.93%) than other groups, which indicated that the yogurt formed a stronger gel structure. The results of the electronic tongue indicated that the bitterness values of group E (−6.12), F (−6.56), and G (−6.27) yogurts were lower than those of group B (−5.12), C (−4.31), and D (−3.79), respectively, which might be attributed to PNS being encapsulated by PWP. The results indicated that PWP-encapsulated PNS could cover the bitterness of PNS and improve the quality of yogurt containing PNS.

## 1. Introduction

Besides its basic nutritional functions, fermented food also has health benefits, e.g., physiological functions, desirable taste, and sensory attributes [1,2,3]. Studies report that the health benefits of yogurt include reducing the risk of Type 2 Diabetes (T2D), alleviating intestinal disease, improving lactose intolerance, and boosting the immune system [4,5,6].

*Panax notoginseng* is widely used in herbal medicine. *Panax notoginseng* saponins (PNSs) are the primary bioactive compounds in *Panax notoginseng* [7]. It has been reported that PNS has a number of beneficial effects, e.g., alleviating osteoporosis and joint antigen-induced arthritis [8], promoting cutaneous wound healing and suppressing scar formation [9], improving hyperlipidemia and alleviating obesity [10], anti-cancer effects [11], as well as protective effects in curing cardiovascular diseases [12].

As a by-product of cheese making, whey was treated as a form of waste in the past [13]. Environmental pollution can be caused by whey due to its high biological oxygen demand and chemical oxygen demand [14]. Whey was then physically processed into different products, e.g., whey powder, whey protein concentrate (WPC), and whey protein isolate (WPI), to reduce environmental pollution [15]. Whey proteins are widely used in foods and pharmaceutics due to their high nutritional value and broad functional properties [16]. It can be employed as a wall material for embedding functional materials due to its film-forming properties. Rose essential oil-encapsulated whey protein concentrate–pectin nanocomplexes were studied [17]. It has been reported that whey protein improves the stability of C-phycocyanin in acidified conditions during light storage [18]. Polymerized whey proteins (PWPs) refer to the soluble whey protein aggregates that are generated at particular temperatures and protein concentrations, which tend to form a gel, though this is not owing to the low salt condition [19]. PWP could improve the functional properties of whey protein, e.g., foaming and emulsification properties [20], and broad their application range. A new type of protein-fortified-set yogurt using PWP as a co-thickener agent was developed [21]. It was reported that the use of PWPs for the microencapsulation of ginsenosides (GSs) effectively masked the bitter taste of fermented milk containing GS [22]. Furthermore, whether encapsulating fish oil with PWP could adequately cover the fishy flavor in yogurts was a question studied by Liu et al. [23]. The PWP-based microencapsulation could effectively mask the bitter taste of tartary buckwheat flavonoids (TBFs) and improve the color of yogurts containing TBF [24].

Food fortified with PNS can be considered functional food [25]. However, the utilization of PNS in functional foods is restricted because of its unpleasant flavor and yellow color. The results of previous studies reveal the characterization of PNS-PWP nanoparticles by encapsulation efficiency, mean particle size, Zeta potential, polydispersity index, fluorescence spectroscopy, thermodynamic properties, Fourier transform infrared spectroscopy, simulated gastrointestinal stability, microstructure, and electronic tongue indicators [26]. When the mass ratio of PNS to PWP was 1:30, the values of the encapsulation efficiency, mean particle size, and zeta potential of PNS-PWP nanoparticles were 92.94%, 55 nm, and −28 mV, respectively, which showed that the nanoparticles had better stability [26]. PNS-PWP nanoparticles not only have the effect of easy preparation and reduction in PNS bitterness but also have the potential to increase the protein content and improve the nutritional value and quality of yogurt when applied to it. This study aimed to investigate the effects of PNS-PWP nanoparticles on the rheological, textural characteristics and bitterness of the yogurt, which could be meaningful for the development of functional foods containing PNS.

## 2. Materials and Methods

### 2.1. Materials

*Panax notoginseng* saponin-polymerized whey (PNS–PWP) nanoparticles were obtained according to our previously reported method [26]. Briefly, whey protein was heated at 80 °C for 30 min to obtain PWP, which was rapidly cooled to room temperature, and then PNS-PWP nanoparticles were obtained by adding PNS powder and stirring for 1 h. Milk and sugar were obtained from a local supermarket (Harbin, China). ABY-8 (*Streptococcus thermophilus*, *L. bulgaricus*, *Bifidobacterium*, *L. acidophilus*) was obtained by Chr. Hansen Co., Ltd. (A/S, Hoersholm, Denmark).

### 2.2. Preparation of Yogurt

According to the results of preliminary experiments, the mass ratio of PNS/PWP at 1:30 was chosen to prepare PNS-PWP nanoparticles. Seven different yogurt formulations were prepared as follows: group A was plain yogurt without PNS or PNS–PWP nanoparticles; groups B, C, and D included yogurt containing 0.8 mg/mL, 1.0 mg/mL, and 1.2 mg/mL free PNS, respectively; groups E, F, and G were yogurt containing 26.68 mg/mL, 33.35 mg/mL, and 40.03 mg/mL of PNS-PWP nanoparticles, respectively. The levels of PNS in groups B, C, and D were the same as those in groups E, F, and G, respectively.

When raw milk was heated to 60 °C using a magnetic agitator (HS 7, Guangzhou, China), the sugar (7%, *w*/*v*) was added slowly and stirred constantly. Then, the milk was heated to 85 °C for 15 min for pasteurization. After pasteurization, the milk was cooled to 41 °C using an ice water bath, and the yogurt starter ABY-8 (0.03%), PNS, or PNS-PWP nanoparticles were added and mixed well. The entire mixture was fermented at 41 °C for 5 h. The yogurt was kept at 4 °C for further use after cooling down.

### 2.3. Rheological Characterization of Yogurt

The Haake Mars 40 rheometer (Thermo Scientific, Waltham, MA, USA) was employed to monitor the process of fermentation for the milk mixture according to the method by Song et al. [27]. Throughout the fermentation process, the entire mixture was kept at a temperature of 41 °C for a duration of 5 min. Afterward, it was tested with mild deformation oscillations at a frequency of 1 Hz and a strain of 0.5% for 5 h. Following fermentation, the oscillation measurement was carried out while the temperature decreased from 41 to 4 °C. Elastic modulus (G′) was determined. After cooling, the oscillation measurement and viscosity of the milk mixture were determined at 25 °C. The shear rate varied from 25 s^−1^ to 500 s^−1^. G′ and viscosity were determined as three replications for each sample.

### 2.4. Texture Profile Analysis

The texture of the yogurt samples was measured using a Texture Analyzer (TA.XT plus, Stable Micro Systems, Godalming, UK) with the TPA mode following the method described by Wang et al. [28] with several alterations. The parameters were set as follows: the type of probe: AB-E/40; speed: 1 mm/s; trigger: 2 g; distance: 15 mm. The hardness, cohesiveness, springiness, gumminess, and chewiness were collected. Each sample was determined by three replications.

### 2.5. Syneresis Measurement

The syneresis of yogurts was determined following the method tested by Córdova-Ramos et al. [29]. In total, 10 g of the yogurt samples were centrifuged at 1200× *g* for 15 min at 4 °C. After centrifugation, the yogurt’s syneresis was equal to the supernatant weight divided by the sample weight and then multiplied by 100%.

### 2.6. pH Values

The pH values of yogurt samples were determined using a pH meter (PHS-3C, Shanghai, China) at room temperature. The pH meter was calibrated through the calibration fluid. The sample was stirred evenly before measurement. Each sample was conducted in triplicates.

### 2.7. Electronic Tongue Analysis

The electronic tongue (SA402B, Atsugi-Shi, Japan) is provided with sensors for umami, bitterness, sourness, saltiness, and astringency. These sensors were utilized to evaluate the taste characteristics of the yogurt. Briefly, the sensors were calibrated, washed, and maintained in the equilibrium position for a duration of 30 s. Then, the yogurt was tested for 30 s. Each sample was repeated three times, and the mean of the three measurement values was utilized for analysis.

### 2.8. Cryo-Scanning Electron Microscopy

The microstructure of the yogurt was observed using cryo-scanning electron microscopy (Xl30 Esem Feg, Eindhoven, The Netherlands) based on the method by Ban et al. [30]. Groups A, D, and G’s yogurts were pre-frozen using liquid nitrogen. The frozen yogurt samples were fractured and etched at −85 °C for 20 min. Images of yogurt samples covered with 300 Å of sputter-coated gold were obtained at 5 kV.

### 2.9. Statistical Analysis

Every experiment was performed three times, and the results were shown as the mean ± standard deviation. Differences among groups were assessed and analyzed using one-way ANOVA and Duncan’s multiple range test with SPSS 20.0 (SPSS Inc., Chicago, IL, USA). A significant threshold of *p* < 0.05 was established.

## 3. Results and Discussion

### 3.1. Rheological Properties of Yogurt during Fermentation

The gelation kinetic profile of yogurt with different levels of free PNS or PNS-PWP nanoparticles during fermentation is displayed in Figure 1. During the whole fermentation process, the gel formation time can be regarded as the first point when G′ > 1 [30]. When G′ < 1, it indicated that all samples were in a liquid-like state. The gelation time of groups A, B, C, D, E, F, and G were 116.2, 123.9, 123.2, 126.7, 84.37, 33.84, and 23.53 min, respectively. The gelation times of groups E, F, and G were shorter than those of groups A, B, C, and D, which indicated that PNS-PWP nanoparticles could reduce the gelation time of yogurt. When the temperature changed from 41 °C to 4 °C, the G′ values of groups A, B, C, D, E, F, and G still increased and reached the maximum value, which suggests the formation of a three-dimensional network gel structure according to the previous study [27]. As shown in Figure 2, The maximum G′ values of groups A, B, C, D, E, F, and G were 3613, 3033, 2877, 2770, 5262, 6519, and 7135 Pa, respectively. No significant difference (*p* > 0.05) in G′ values among groups E, F, and G were found, but their G′ values were significantly higher than those without PNS-PWP nanoparticles. These results suggest that group G is favorable for gelation during fermentation.

After the cooling oscillation measurement, the apparent viscosity of groups A to G is presented in Figure 3. Yogurt viscosity varies depending on the type of milk, protein content, and total solids content [31]. It can be observed that the viscosity of groups A to G decreased with the increasing shear rate and eventually showed a constant viscosity, which might be related to the breakup of the aggregates [32]. The viscosity of groups B, C, and D continued decreasing and was smaller than that of group A. This was associated with the addition of PNS. Groups E, F, and G showed higher viscosity than the other groups and increased with the increasing PNS–PWP nanoparticle levels. This could be explained by the larger complex formed by the interaction between PWP and casein micelles [33]. A similar result was reported where encapsulated fish oil with PWP samples could improve the viscosity of yogurt [23].

### 3.2. Texture Profile Analysis

The texture of food products can represent all the rheological and structural attributes perceptible by means of mechanical, tactile, visual, and auditory receptors [34]. Texture is also one of the most important indicators affecting the quality of yogurt. Figure 4 shows the results of the hardness, springiness, cohesiveness, gumminess, and chewiness of the yogurt with different levels of free PNS or PNS–PWP nanoparticles after fermentation.

A significant difference (*p* < 0.05) was observed in the hardness of different yogurt groups. Group G showed the highest hardness value (506 g). As the PNS level increased from 0.8 to 1.2 mg/mL, the hardness value decreased significantly (*p* < 0.05) from 218 g in group B to 153 g in group D. These results could be attributed to the interaction of casein in milk with the PNS. This result was analogous to that reported by Sun et al. [24], who found that the hardness value of free tartary buckwheat flavonoids in yogurt decreased compared with the control yogurt. But the hardness of groups E to G increased significantly (*p* < 0.05) from 266 to 506 g compared with group A (202 g). This might be because a higher protein content would cause a higher degree of cross-linkage in the gel network, resulting in a much denser and more rigid gel structure [35].

A free PNS in groups B, C, and D might affect the texture of the yogurt and lead to a decrease in springiness compared with group A. Group G exhibited the highest springiness (0.87) compared to the other groups. Sun et al. [24] reported similar results where tartary buckwheat flavonoids encapsulated by PWP improved the springiness of yogurt. Except for group G (0.42), the cohesiveness values of groups B to F were lower than that of group A (0.30). Since cohesiveness could indicate the strength of the internal bonds in the yogurt’s structure [36], the higher cohesiveness value of group G indicated a stronger structure compared with the other groups. For gumminess, the E (66), F (82), and G (212) groups had significant (*p* < 0.05) differences and were higher than that of groups A (61), B (57), C (51), and D (45). Significant (*p* < 0.05) differences in the chewiness of groups E, F, and G were observed. The chewiness values of groups E, F, and G were higher than the other groups, which indicates that PNS-PWP nanoparticles might improve the structure of yogurt. These results indicate that group G had a positive impact on the texture of the yogurt.

### 3.3. Syneresis

Syneresis refers to the contraction of the gel of set yogurt, which can cause whey separation [37]. Whey separation means the presence of whey appearing on the gel’s surface [31]. The presence of wheying-off makes consumers assume there is a microbiological problem with the product, which has an adverse impact on consumer acceptance of products [38]. Figure 5 shows the syneresis values of groups A to G at 8.37 ± 0.64%, 10.87 ± 0.59%, 11.23 ± 1.21%, 11.80 ± 0.10%, 10.57 ± 0.68%, 9.91 ± 0.56%, and 6.93 ± 0.31%, respectively. It suggests that the addition of free PNS increased the syneresis of the yogurt. No significant (*p* > 0.05) difference in syneresis was observed among groups B, C, and D. While the differences among groups A, D, and G were significant, which indicates that the addition of PNS-PWP nanoparticles might improve the quality of yogurt by reducing syneresis, this might be ascribed to the formation of a denser gel microstructure. It was reported that the addition of dried dairy ingredients resulted in a reduction in syneresis in yogurt [21]. Group G exhibited the lowest syneresis value, which suggests it might have better sensory performance.

### 3.4. Changes in pH

The pH values of groups A to G are presented in Figure 6. No significant (*p* > 0.05) difference among groups B, C, and D was observed. Groups B, C, and D exhibited lower pH values than the other groups. According to the results reported by He et al. [39], this might be due to the positive effect of PNS during fermentation. Wang et al. [22] reported similar results in which ginsenosides available for probiotics resulted in a pH decrease during fermentation compared to the control yogurt. No significant (*p* > 0.05) difference between group G (4.47 ± 0.02) and A (4.49 ± 0.01) was found. This might be caused by PNS being encapsulated in PWP nanoparticles.

### 3.5. Bitterness by Electronic Tongue

The electronic tongue can convert electrical impulses into taste signals and measure the taste of yogurt, which can remove subjectivity from sensory evaluation [40]. The bitterness values are shown in Figure 7. The bitterness values of the yogurt increased from −5.12 ± 1.77 to −3.79 ± 0.78 as the free PNS content increased from 0.8 mg/mL to 1.2 mg/mL. Groups E, F, and G showed bitterness values of −6.12 ± 0.43, −6.56 ± 0.15, and −6.27 ± 0.34, respectively. It indicates that the bitterness values of yogurt decreased significantly (*p* < 0.05) after adding PNS-PWP nanoparticles and had no significant difference compared with group A (−6.62 ± 0.06). The preparation of PNS-PWP nanoparticles using PWP as a wall material for yogurt application not only increases the protein content and reduces the bitter taste of PNS to improve the quality of yogurt but may also increase the bioavailability of PNS due to the protection of PWP in gastric juice against the degradation of PNS.

### 3.6. Microstructure

Figure 8 shows the cryo-SEM of groups A, D, and G. Casein micelles and open cavities were observed in all samples. In comparison with group A, group D showed larger open cavities and looser gel networks, resulting in increased syneresis. Group G formed a relatively small open cavity and a denser network structure. This might be due to the fact that the combination of PWP and casein micelles leads to a bridge forming between casein particles and the formation of a network structure, which improves the consistency and increases the water retention of yogurt [33]. Studies reported that the bacterially acidified cold-set gelation of pre-polymerized whey proteins may be a novel method to improve the texture and water-binding properties of fermented dairy foods, such as yogurt [31].

## 4. Conclusions

Yogurts with different levels of free PNS or PWP-PNS nanoparticles were formulated. The yogurt sample containing 40.03 mg/mL of PNS-PWP nanoparticles showed a shorter gelation time, highest elastic modulus, higher apparent viscosity, higher hardness, and the lowest syneresis. In comparison with yogurt containing free PNS, yogurt with PNS–PWP nanoparticles had a lower bitter taste. Polymerized whey protein encapsulation may be a useful technology for formulations of yogurt products fortified with functional ingredients to give a special taste. The stability of yogurt with PNS-PWP nanoparticles during storage deserves further investigation.

## Figures and Tables

**Figure 1 foods-13-00486-f001:**
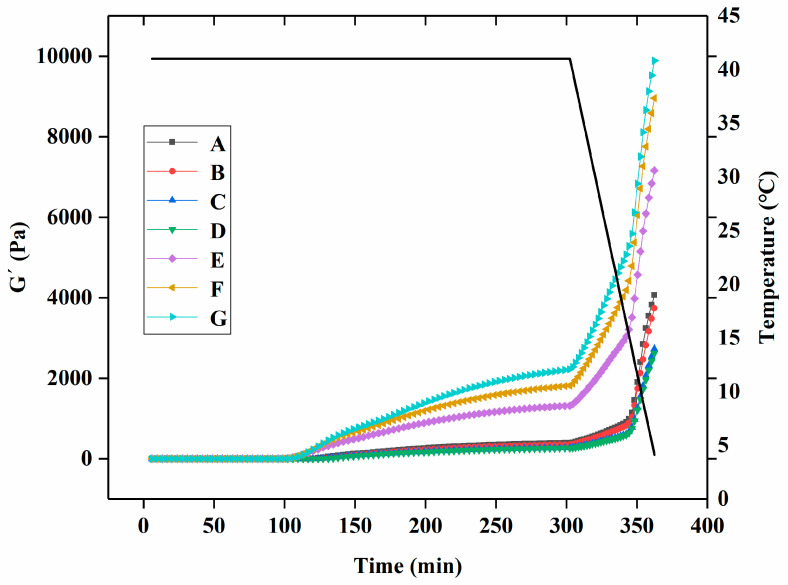
Rheological parameters of groups A, B, C, D, E, F, and G’s yogurts. A (without free PNS and PNS-PWP nanoparticles); B (0.8 mg/mL free PNS); C (1.0 mg/mL free PNS); D (1.2 mg/mL free PNS); E (26.68 mg/mL PNS-PWP nanoparticles); F (33.35 mg/mL PNS-PWP nanoparticles); and G (40.03 mg/mL PNS-PWP nanoparticles).

**Figure 2 foods-13-00486-f002:**
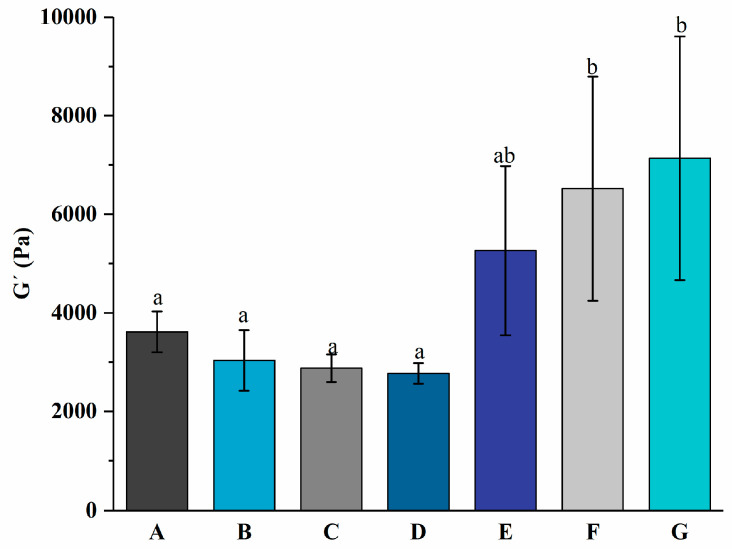
Maximum elastic modulus of groups A, B, C, D, E, F, and G’s yogurts during fermentation. A (without free PNS and PNS-PWP nanoparticles); B (0.8 mg/mL free PNS); C (1.0 mg/mL free PNS); D (1.2 mg/mL free PNS); E (26.68 mg/mL PNS-PWP nanoparticles); F (33.35 mg/mL PNS-PWP nanoparticles); and G (40.03 mg/mL PNS-PWP nanoparticles). Different subscript letters indicate a significant difference (*p* < 0.05).

**Figure 3 foods-13-00486-f003:**
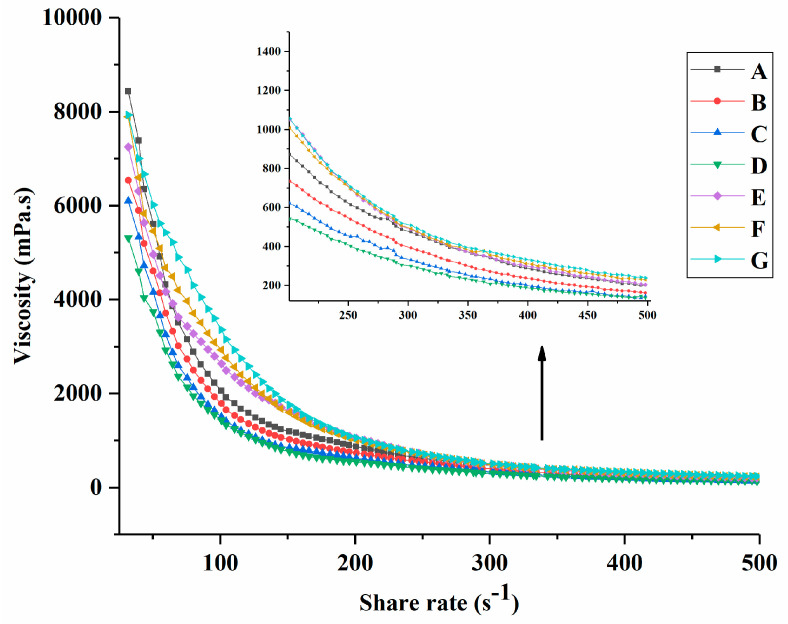
The apparent viscosity of groups A, B, C, D, E, F, and G’s yogurts. A (without free PNS and PNS-PWP nanoparticles); B (0.8 mg/mL free PNS); C (1.0 mg/mL free PNS); D (1.2 mg/mL free PNS); E (26.68 mg/mL PNS-PWP nanoparticles); F (33.35 mg/mL PNS-PWP nanoparticles); and G (40.03 mg/mL PNS-PWP nanoparticles).

**Figure 4 foods-13-00486-f004:**
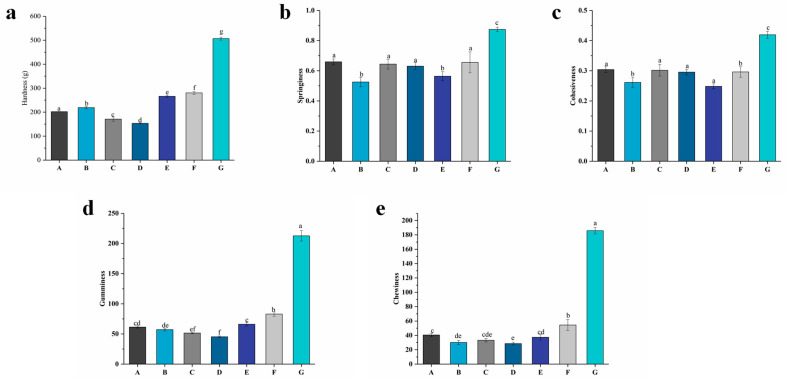
Textural properties of groups A, B, C, D, E, F, and G’s yogurts. (**a**): Hardness of yogurt; (**b**): springiness of yogurt; (**c**): cohesiveness of yogurt; (**d**): gumminess of yogurt; (**e**): chewiness of yogurt. A (without free PNS and PNS-PWP nanoparticles); B (0.8 mg/mL free PNS); C (1.0 mg/mL free PNS); D (1.2 mg/mL free PNS); E (26.68 mg/mL PNS-PWP nanoparticles); F (33.35 mg/mL PNS-PWP nanoparticles); and G (40.03 mg/mL PNS-PWP nanoparticles). Different subscript letters indicate a significant difference (*p* < 0.05).

**Figure 5 foods-13-00486-f005:**
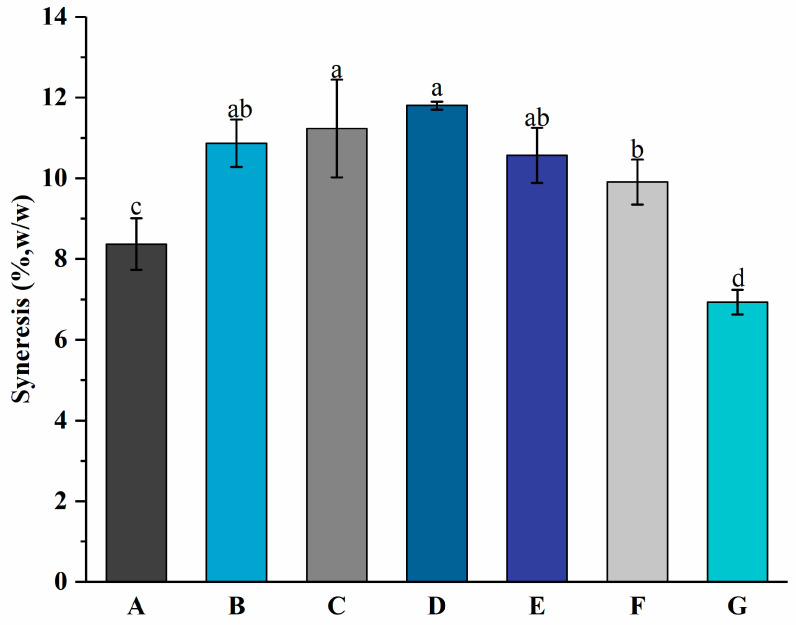
Syneresis of groups A, B, C, D, E, F, and G’s yogurts. A (without free PNS and PNS-PWP nanoparticles); B (0.8 mg/mL free PNS); C (1.0 mg/mL free PNS); D (1.2 mg/mL free PNS); E (26.68 mg/mL PNS-PWP nanoparticles); F (33.35 mg/mL PNS-PWP nanoparticles); and G (40.03 mg/mL PNS-PWP nanoparticles). Different subscript letters indicate a significant difference (*p* < 0.05).

**Figure 6 foods-13-00486-f006:**
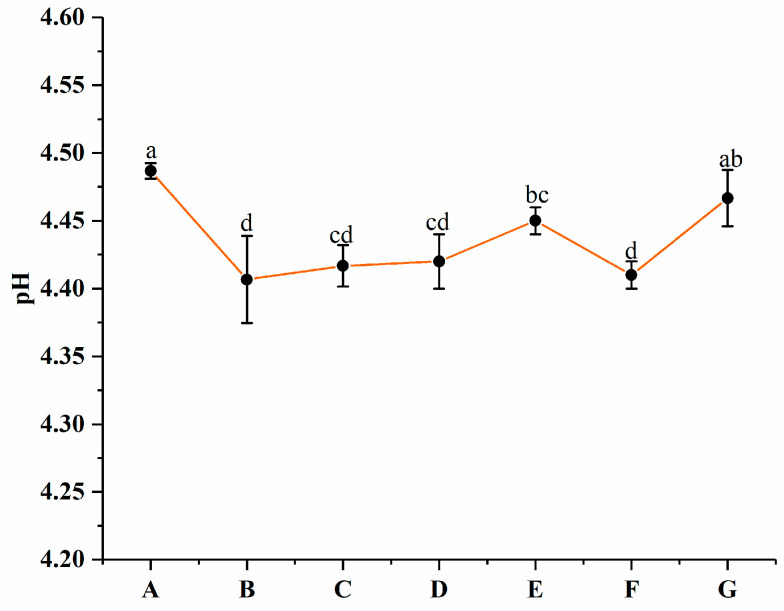
pH values of groups A, B, C, D, E, F, and G’s yogurts. A (without free PNS and PNS-PWP nanoparticles); B (0.8 mg/mL free PNS); C (1.0 mg/mL free PNS); D (1.2 mg/mL free PNS); E (26.68 mg/mL PNS-PWP nanoparticles); F (33.35 mg/mL PNS-PWP nanoparticles); and G (40.03 mg/mL PNS-PWP nanoparticles). Different subscript letters indicate a significant difference (*p* < 0.05).

**Figure 7 foods-13-00486-f007:**
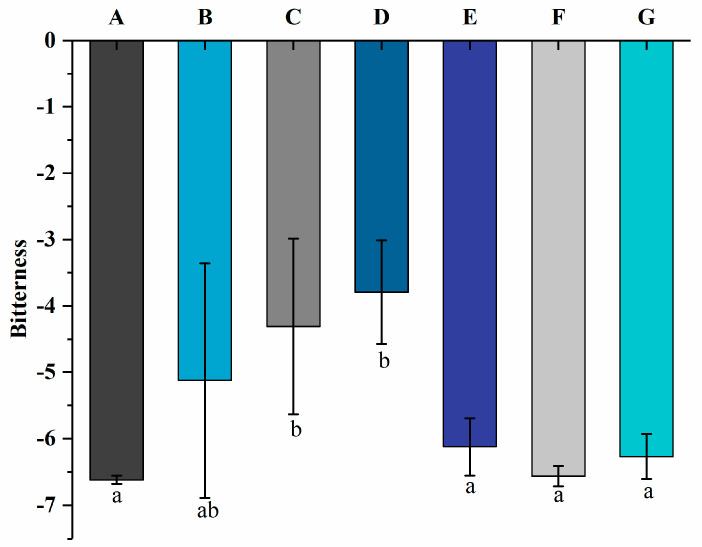
The bitterness values of groups A, B, C, D, E, F, and G’s yogurts using an electronic tongue. A (without free PNS and PNS-PWP nanoparticles); B (0.8 mg/mL free PNS); C (1.0 mg/mL free PNS); D (1.2 mg/mL free PNS); E (26.68 mg/mL PNS-PWP nanoparticles); F (33.35 mg/mL PNS-PWP nanoparticles); and G (40.03 mg/mL PNS-PWP nanoparticles). Different subscript letters indicate a significant difference (*p* < 0.05).

**Figure 8 foods-13-00486-f008:**
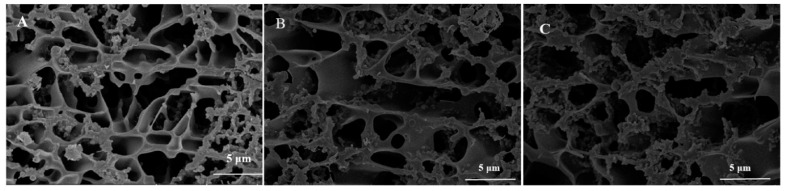
Cryo-SEM photographs of groups A, D, and G’s yogurts. (**A**) (without free PNS and PNS-PWP nanoparticles); (**B**) (1.2 mg/mL free PNS); and (**C**) (40.03 mg/mL PNS-PWP nanoparticles).

## Data Availability

Data is contained within the article.

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
