# Peer review of "Effects of Panax notoginseng Saponins Encapsulated by Polymerized Whey Protein on the Rheological, Textural and Bitterness Characteristics of Yogurt"

_foods, 2024, doi:10.3390/foods13030486_

Round 1

Reviewer 1 Report

Comments and Suggestions for Authors

-Please provide a concise description or additional details regarding the synthesis method used to obtain the Panax Notoginseng Saponins-Polymerized Whey (PNS-PWP) nanoparticles mentioned in section 2.1?

-Provide insights into encapsulation efficiency, particle size, stability assessment of PNS-PWP nanoparticles, and the methodology used for characterization.

-Consider comparing these findings with previous studies that explore encapsulation techniques or similar functional ingredient additions to yogurt. Highlight the uniqueness or advantages of PNS-PWP nanoparticles in comparison to other encapsulation methods.

-Line 86- start with capital after the period “well. the mixture”

-Line 159- please fix the sentence “varies aredepending”

-Please explain the reason for the lower pH value for F (33.35 mg/mL PNS-PWP nanoparticles) treated yogurt while E and G-treated yogurt showed higher pH.

-Line 264- please separate “formsa”

-A comprehensive discussion on each parameter's implications could improve the paper's depth and relevance.

-Expand statistical analysis and discuss significance where applicable.

-Elaborate on the mechanisms behind the observed changes in rheological and textural characteristics.

-Include information on the long-term stability of yogurt fortified with PNS-PWP nanoparticles. Assess changes in properties over an extended storage period to gauge the robustness of the encapsulation and its impact on yogurt quality.

-Conduct a sensory evaluation involving consumers to assess preferences and acceptability of yogurts with PNS or PNS-PWP nanoparticles. This could provide valuable insights into the consumer market and product acceptability.

-Discuss potential health benefits or concerns associated with the consumption of yogurts fortified with PNS-PWP nanoparticles. Address the bioavailability, safety, and effectiveness of PNS in this encapsulated form.

-To enhance the conclusion, it'd be valuable for the authors to discuss the practical implications of these findings for consumers or the market. Also, suggesting future research areas or acknowledging study limitations would provide a more comprehensive ending.

-Strengthen the conclusion by summarizing key findings and discussing broader implications.

Author Response

Dear Reviewer,

Reviewer 2 Report

Comments and Suggestions for Authors

Line 60 This statement requires a reference “Food fortified with PNS could be considered as a functional food

Line 70 I would suggest to avoid the usage of personal pronouns like “our” and just leave “previously reported method”.

Line 75 It is not clear what “the results of preliminary experiments” are. Please clarify.

Line 83 Please clarify why it was decided to add sugar to milk before the fermentation?

Line 175 Please correct the articles in the beginning of the sentence.

Overall, the results of the paper are interesting. I suppose it may also be of interest to conduct a sensory panel of yoghurt samples to assess whether even the lowered level of bitterness is acceptable for commercial yoghurt production.

Comments on the Quality of English Language

There are some typos in the text, but in general the quality of the English is high.

Author Response

Dear Reviewer,
